# An In Vitro Approach to Studying the Microbial Community and Impact of Pre and Probiotics under Anorexia Nervosa Related Dietary Restrictions

**DOI:** 10.3390/nu13124447

**Published:** 2021-12-13

**Authors:** Litai Liu, Carlos Poveda, Paul E. Jenkins, Gemma E. Walton

**Affiliations:** 1Department of Food and Nutritional Sciences, University of Reading, Reading RG6 6AX, UK; l.liu2@pgr.reading.ac.uk (L.L.); c.g.povedaturrado@reading.ac.uk (C.P.); 2School of Psychology and Clinical Language Sciences, University of Reading, Reading RG6 6ES, UK; pej106@gmail.com

**Keywords:** anorexia nervosa, gut-brain axis, gut microbiota, neurotransmitters, psychobiotics, prebiotics, probiotics

## Abstract

Individuals with anorexia nervosa (AN) often suffer psychological and gastrointestinal problems consistent with a dysregulated gut microbial community. Psychobiotics have been postulated to modify microbiota and improve mental well-being and gut symptoms, but there is currently a lack of evidence for such approaches in AN. The aim of this study was to use an in vitro colonic model to evaluate the impact of dietary restrictions associated with AN on the intestinal ecosystem and to assess the impact of pre and probiotic intervention. Bacteriology was quantified using flow cytometry combined with fluorescence in situ hybridisation and metabolic end products (including neurotransmitters) by gas chromatography and liquid chromatography mass spectrometry Consistent with previous research, the nutritional changes significantly reduced total microbiota and metabolites compared with healthy conditions. Pre and probiotic supplementation on restricted conditions enhanced the microbial community and modulated metabolic activity to resemble that of a healthy diet. The model system indicates that nutritional changes associated with AN can impact the microbial community, and that these changes can, at least in part, be restored through the use of pre and probiotic interventions.

## 1. Introduction

Anorexia nervosa (AN) is a serious psychiatric disorder characterised by restriction of dietary intake (typically leading to low body weight) and a distorted perception of body shape [1]. Compared to those without eating disorders (EDs), significant differences in gut microbiota composition have been demonstrated in individuals with AN, with an imbalanced relative abundance in Gram-positive/Gram-negative bacteria [2,3,4,5]. Most commensal bacterial species in gut microbiota are members of the Bacteroidetes and Firmicutes phyla. The F/B ratio is high in obese people compared to lean people, and tends to reduce with weight loss [6,7]; therefore, the F/B ratio could be impacted by the presence of AN symptoms [2,5]. Individuals with AN have been observed to have decreased levels of SCFA-producing *Roseburia* [2,5], and a lower abundance of *Lactobacillus* compared to healthy volunteers [8,9]. The number of intestinal bacteria in individuals with AN is severely reduced because of the limited dietary intake, leading to a gut microbial community adapting to a starvation state [4,10]. Indeed, reduced levels of microbial growth in a nutrient-limited environment is well-known [11]. However, such an alteration could be of importance when considering dietary restrictions in AN.

The gut microbial community has important interactions with host metabolism, impacting on body weight regulation and hormonal processes, along with a direct impact on the brain and behaviour via the gut-brain axis (GBA) [12,13,14,15,16]. Research indicates a key role of gut microbiota in the regulation of behaviour, mood, gastrointestinal symptoms, nutrient metabolism, and satiety and appetite; functions often altered in AN [17]. Within AN, changes in nutritional intake result in an altered microbial community that could impact GBA communication and further affect neurological function. This opens up the prospect of studying how the gut microbiota impact the brain and whether modification of gut microbiota could be helpful in the fight against AN [18].

Gut microbiota produce short-chain fatty acids (SCFA) via saccharide fermentation. SCFA could be important molecules in AN as increased SCFA levels in faecal samples have been observed in obese and overweight people compared to AN sufferers [4,19,20], and could relate to physiological processes including the GBA and immune system [21,22,23]. Furthermore, SCFA act as signalling molecules and are involved in regulating gut transit time, appetite and energy homeostasis [23]. Additionally, microorganisms have been reported to produce, or be involved in the production of, various neuroactive compounds such as serotonin (5-HT), dopamine (DA), and norepinephrine (NE) [24]. 5-HT is one of the main neurotransmitters located in the brain, but almost 90–95% of total body 5-HT is located in the gastrointestinal tract (GIT) and is secreted via the epithelial enterochromaffin cells (ECs) of the gut, and thus is involved in GBA bidirectional signalling [25]. A study by Yano et al. [26] indicated that enteric bacteria, such as *Streptococcus, Escherichia* and *Enterococcus* species may modulate host 5-HT biosynthesis by increasing its precursor, tryptophan, in plasma. Reduced tryptophan has been associated with anxiety and borderline personality symptoms in patients with AN and bulimia nervosa [27,28]. However, serotonergic activity status can be predicated from 5-HT metabolites 5-HIAA, and long-term weight-restored patients with AN have elevated 5-HIAA in cerebrospinal fluid [29].

Psychobiotics are supplements designed to modulate the gut microbiota to enhance mood and can be in the form of beneficial bacterial supplements (probiotics) or support for positive bacteria already within the gut (prebiotics). Ultimately, these must also affect bacteria–brain communication [30]. The bacteria enhanced through these interventions may therefore result in elevated levels of neuroactive compounds such as 5-HT and gamma-aminobutyric acid (GABA), which act on the GBA [31,32]. The bacteria most commonly utilised as psychobiotics are probiotics that have shown potential effects on psychological and physiological conditions such as improving anxiety, depression and appetite levels [33,34]. Few studies have examined the psychophysiological impact of prebiotics. Soluble fibre fructo-oligosaccharides (FOS) are a nutritional source for *Bifidobacterium*, reflected by their bifidogenic capability, thus they promote its activity and proliferation in the gut [34]. It is unknown how an ‘anorexic’ dietary patterns might affect the microbiota, and further how pre and probiotics may impact this.

In vitro models can be a useful tool for determining how microbial communities grow in the presence of different nutrients with physiologically-relevant conditions including human large intestine nutrients, temperature and pH, but without the need for human participants. A three-stage continuous culture system was developed by Macfarlane et al. [35] to mimic nutritional and physiochemical conditions of microbiota in the colon. The constituents of culture medium were determined on the basis of large intestine contents with the main substrates available for microorganisms determined to be dietary fibre, proteins, oligosaccharides and peptides that evade digestion in the small intestine, as well as a variety of host-derived substances, including mucins, pancreatic secretions and exfoliated epithelial cells [36,37]. In the current study a media approximated to anorexic dietary intake, along with host-derived substances, was developed for use in a three-stage continuous culture system, enabling the activities of intestinal bacteria to be established under these nutrient-deficit conditions. The subsequent impact of pre and probiotics on this community will be investigated.

## 2. Materials and Methods

### 2.1. Three-Stage Compound Continuous Culture System

A three-stage continuous culture system was set up in sequence simulating the proximal (V1, 80 mL, pH = 5.5), transverse (V2, 100 mL, pH = 6.2) and distal colon (V3, 120 mL, pH = 6.8) comprised of a scaled-down version of 3 regions of the GI tract (Figure 1). The system was maintained in anaerobic conditions by supplying N_2_ (15 mL/min) and holding a continuous temperature of 37 °C via a circulating water bath. The systems were inoculated to give a final concentration of 6% faecal slurry; samples were donated from 4 healthy female donors (age range 25–43 years). The faecal donors had not taken antibiotics within 6 months of the experiment and were not regular consumers of prebiotic or probiotic supplements. Collected faecal samples were placed in anaerobic jars (AnaeroJarTM 2.5L, Basingstoke, UK, Oxoid Ltd.) with anaerobic sachets (AnaeroGen, Oxoid) and used within 2 h of production. To prepare the faecal sample, a 1 in 5 (wt:v) faecal slurry with PBS (anaerobic phosphate buffered saline; 0.1 mol/L; pH 7.4) was homogenised in a stomacher (Stomacher 400, Seward, West Sussex, UK) for 2 min (240 paddle beats/min).

The experiment was conducted 4 times with a different faecal donor for each run. Following inoculation, the system was run for 24 h to allow the bacteria to multiply within the vessels. After this, the flow was started with a retention time appropriate to anorexic patients [38] when considering the operating volume (300 mL) and anorexic retention time (64 h, flow rate 4.68 mL/h) of the gut model system. The first steady state (SS1), when equilibrium was reached, was after 512 h (8 full volume turnovers) using standard gut model media (HC feeding). A second steady state (SS2) was achieved after a further 512 h, using nutrient-restricted media (AN feeding). This was determined by assessing the stability of the SCFA over 3 consecutive days. Restricted media continued to be used along with a potential psychobiotic, FOS (1.67 g/daily) or *Saccharomyces boulardii* (5 × 10^8^ cfu) treatment daily into V1 for at least 512 h upon which third steady state (SS3) was achieved. Samples were collected at three time points (SS1, SS2, and SS3).

### 2.2. Gut Model Medium Determination in HC, AN, and AN with Pre and Probiotic Feeding

The AN gut model system and the medium concentration was modelled on the gut environment conditions of AN individuals. Jauregui et al. and Raatz et al. [39,40] outlined the typical dietary intake of restrictive AN patients compared to healthy people. By considering both healthy and restrictive diets and the nutrient constituents of gut model media (Table 1) likely to reach the colon were determined [35]. In terms of other components found within the media representative of human secretions, mucin and bile salts, were included at the same concentration in both media.

### 2.3. Treatments

The probiotic yeast *Saccharomyces boulardii* (OptiBac Probiotics Ltd., Hampshire, UK) product contained 5 × 10^9^ cfu live culture powder in each capsule. The prebiotic FOS (Orafti ^®^ P95) was obtained from BENEO (Orafti ^®^ P95, Tienen, Belgium).

### 2.4. Preparation of the Samples for SCFA/BCFA Analysis, Neurotransmitter Analysis and Bacterial Community Analysis

Samples were taken at SS1, SS2 and SS3 time points from proximal, transverse and distal vessels. 1 mL of gut model fluid was centrifuged in a micro centrifuge Eppendorf tube (1.5 mL) at 13,000× *g* for 10 min and the supernatant was stored at −20 °C prior to SCFA/BCFA analysis. A further 0.4 mL was collected and centrifuged at 13,000× *g* for 10 min then stored at −20 °C for neurotransmitter analysis.

For bacterial community analysis, a 750 μL supernatant of gut model fluid was centrifuged at 13,000× *g* for 5 min. The pellet was then resuspended in 375 μL filtered 0.1 M PBS and fixed by 1125 μL filtered paraformaldehyde (PFA 4% *v*/*v*) for 4 to 8 h at 4 °C. The sample was washed twice with 1 mL PBS to remove PFA and resuspended in filtered 600 μL ethanol-PBS (1:1, *v*/*v*). The samples were kept at −20 °C prior to FISH analysis.

### 2.5. In Vitro Enumeration of Bacterial Population by Flow-Fluorescent In Situ Hybridisation (FISH)

The bacterial population was analysed using fluorescent in situ hybridisation coupled to flow cytometry (BD Accuri ^TM^ C6 Plus, Basingstoke, United Kingdom), detecting at 488 nm and 640 nm and analysed using Accuri CFlow Sampler software. Samples were removed from storage at −20 °C. After defrosting and vortexing for 10 s, permeabilisation steps were conducted using 500 μL 0.1 M PBS added to 75 μL fixed samples and centrifuged at 13,000× *g* for 3 min. The pellets were resuspended in 100 μL of TE-FISH buffer (Tris-HCl 1 M pH 8, EDTA 0.5 M pH 8, filtered distilled water, 0.22 μm pore size filter with the percentage of 10:10:80) containing lysozyme solution (1 mg/mL of 50,000 U/mg protein) and incubated for 10 min in the dark at room temperature and then centrifuged at 13,000× *g* for 3 min. Pellets were washed with 0.1 M 500 μL PBS and then washed with 150 μL hybridisation buffer (0.9 M NaCl, 0.2 M Tris/HCl pH 8.0, 30% formamide, ddH_2_O, 0.01% sodium dodecyl sulphate) and centrifuged at 13,000× *g* for 3 min. Pellets were then resuspended in 1 mL of hybridisation buffer, homogenised, and 50 μL with 4 μL of different probes aliquoted into Eppendorf tubes (1.5 mL) were incubated at 36 °C overnight. Differences in bacterial populations were quantified with oligonucleotide probes aimed to target specific regions of 16S rRNA. The individual probes used (Eurofins, Wolverhampton, UK) in this study are shown in Table 2. Non-EUB and EUB338-I-II-III were linked to fluorescence Alexa 488 at the 5′ end, and group-specific probes were linked to fluorescence Alexa 647. Non-EUB and EUB338 were linked to Alexa 647 at the 5′ end as controls to adjust threshold. 4 μL of EUB338-I-II-III was added together with 4 μL specific probes. 125 μL of hybridisation buffer was added to each Eppendorf tube, after incubation samples were vortexed and centrifuged (13,000× *g*, 3 min). Supernatants were removed and pellets were washed with 175 μL washing buffer solution (0.064 M NaCl, 0.02M Tris-HCl (pH 8.0), 0.5 M EDTA (pH 8.0), 0.01% sodium dodecyl sulphate, 956.2 μL of ddh_2_O), vortexed and incubated at 38 °C in a heating block for 20 min to remove non-specific binding of the probe. Afterwards samples were centrifuged (13,000× *g*, 3 min) and supernatants removed. Pellets were resuspended in an appropriate volume of PBS on the basis of flow cytometry (FCM) load. Number of bacteria were then calculated through determination of FCM reading and PBS dilution.

### 2.6. Neurotransmitter Analysis by Liquid Chromatography Mass Spectrometry (LCMS)

#### 2.6.1. Reagents and Chemicals

HPLC Plus grade acetonitrile (≥99.9%) was purchased from Sigma-Aldrich (Kent, UK). Formic acid (≥99% LC/MS grade, HiPerSolv CHROMANORM ^®^) was purchased from VWR. Centrifuge tube filter (Corning ^®^ Costar ^®^ Spin-X ^®^, 0.22 μm Pore CA Membrane, Sterile, 96/Case, Polypropylene) was purchased from Sigma-Aldrich, which was used to filter gut model fluid samples. Analytical standards powder including LC-MS grade dopamine hydrochloride (99%) and L (-)-Epinephrine (99%) were purchased from Alfa Aesar (Lancashire, UK). L-Noradrenaline (98%), Gamma-Aminobutyric acid (99%) and serotonin were purchased from Sigma-Aldrich Co Ltd.

#### 2.6.2. Stock Solutions, Calibration Standards and Sample Preparation

Separate standard stock solutions (10000 ng/mL) of 5 analytes, including 5-HT, DA, GABA, NE and Epinephrine (EPI) were individually prepared in HPLC water. A 1000 ng/mL mixed standard solution containing the 5 analytes was made by acquiring aliquots of each separate stock solution. The mixed standard solution was appropriately diluted with HPLC water to prepare a calibration series. A calibration series of spiked standard samples was prepared including 10 levels: 1, 10, 50, 100, 250, 500, 750, 1000, 2500, and 5000 ng/mL. Samples were removed from storage at −20 °C. A 400 µL sample of gut model fluid supernatant was collected in a centrifuge tube filter (Sigma-Aldrich, 0.22 μm, Polypropylene) and then centrifuged at 13,000× *g* for 10 min at 4 °C (SANYO MSE Mistral 3000i, Sanyo Gallenkap PLC, UK) and the supernatant remained. 200 µL of HPLC water (Blank), calibration standard samples and gut model samples were placed in 96-well plates.

#### 2.6.3. LCMS System

Samples were measured using online Nexera LC System coupled to LCMS-8050 triple quadrupole (QQQ) mass spectrometry (Shimadzu Corporation, Kyoto, Japan). Data were processed using LabSolutions LCMS version 5.65 software.

#### 2.6.4. Liquid Chromatography (LC) Conditions

The chromatographic separation of analytes was obtained from Discovery HS F5-3 column (2.1 mmI.D. × 150 mmL. 3 μm particle size, Sigma-Aldrich Co Ltd., P/N 567503-U). The mobile phase consisted of 0.1% formic acid in water (mobile A) and 0.1% formic acid in acetonitrile (mobile B). For the entire analysis, the flow rates of both mobile phases were 0.25 mL/min, and the autosampler temperature at a maintained constant temperature was set to 4 °C. The gradient elution programs were as follows: B conc. 25% (5 min) → 35% (11min) → 95% (15 min) → 95% (20 min) → 0% (20.01–25 min).

#### 2.6.5. Mass Spectrometry (MS) Conditions

The LC/MS-8050 triple quadrupole (QQQ) detector was operated in the multiple reaction monitoring (MRM) mode using the polarity-switching electrospray ionisation (ESI) mode. The optimal conditions were as follows: dry gas temperature was 300 °C, dry gas flow rate of 10.0 L/min. 4 µL samples were injected. Samples were measured as the target compounds based on MRM. For the analysis of primary metabolites 5-HT, DA, GABA, NE and EPI, LC/MS Method Package for Primary Metabolites (Shimadzu Corporation, Kyoto, Japan) was used. The MRM transitions of the native, stable isotopes, retention times and other conditions are shown in Table 3.

#### 2.6.6. Quantification of Samples

A linear calibration curve was generated based on the detected signal proportional to the concentration of the analyte. Briefly, result validation was performed following published procedures [56]. Good linearity with R^2^ greater than 0.98 was obtained across the set calibration in the range from 1 ng/mL to 5000 ng/mL for each of the analytes, with accuracy within 100% ± 20%. Quantification of samples was determined by calibration with 5 analytes including 5-HT, DA, EPI, GABA and NE.

### 2.7. Short Chain and Branched Chain Fatty Acid Analysis by Gas Chromatography

The concentration of SCFA was determined by Gas chromatography (GC) as previously described by Richardson et al. [57]. Individual solution standards at 5 mM were prepared for acetate, iso-butyrate, butyrate, propionate, valerate, iso-valerate and lactate. The external standard solution contained acetate (30 mM), iso-butyrate (5 mM), n-butyrate (20 mM), propionate (20 mM), n-valerate (5 mM), iso-valerate (5 mM) and lactate (10 mM). 1 mL of each sample was vortexed and transferred into a flat-bottomed glass tube (100 mm × 16 mm, Fisher Scientific UK Ltd., Loughborough, UK) with 0.5 mL concentrated HCl, 50 μL of 2-ethylbutyric acid (0.1 M internal standard solution, Sigma, Poole, UK) and 2 mL diethyl ether. Samples were vortexed for 1 min at 1500 rpm and then centrifuged (2000× *g*, 10 min, 4 °C, SANYO MSE Mistral 3000i, Sanyo Gallenkap PLC, UK). 2 mL of diethyl ether top layer and 50 μL of N-(tert-butyldimethylsi lyl)-N-methyltrifluoroacetamide (MTBSTFA; Sigma-Aldrich, Poole, UK) were added into a GC screw-cap vial. Samples were kept at room temperature for 72 h to enable complete derivatisation prior to GC analysis. A GC Agilent 7890B gas chromatograph (Agilent, Cheshire, UK) using an HP-5ms (L × I.D. 30 m × 0.25 mm, 0.25 μm film thickness) coating of crosslinked (5%-phenyl)-methylpolysiloxane (Hewlett Packard, UK) was used for SCFA detection. 1 μL of each sample was injected with a run time of 17.7 min. Injector and detector temperatures were 275 °C and the column temperature programmed from 63 °C to 190 °C by 5 °C and held at 190 °C for 30 min. Helium was the carrier gas (flow rate, 1.7 mL/min, head pressure, 133 KPa). Peak areas were integrated using Agilent Chemstation software (Agilent Technologies, Basingstoke, UK). SCFA production was quantified by single-point internal standard method as described by Liu et al. [58]. Peak areas of the standard (acetate, butyrate, propionate, valerate, iso-valerate and iso-butyrate) were used to calculate the response factors for each organic acid with respect to the internal standard.

### 2.8. Statistical Analysis

Data from FCM-FISH, LC/MS and GC were analysed with SPSS version 27 (IBM Corp., Armonk, NY, USA). Changes in specific bacterial groups, neurotransmitters and SCFA/ BCFA production were assessed between the 3 steady states using a one-way analysis of variance (ANOVA). Significant differences were assessed by post hoc Tukey HSD (Honestly Significant Difference) test. Statistical significance was set *p* < 0.05. Analyses were performed using GraphPad Prism 9.0 (GraphPad Software, La Jolla, CA, USA).

## 3. Results

### 3.1. Bacterial Enumeration

Changes in bacterial compositions in the gut model systems are reported in Figure 2. Compared to the HC-media (SS1), there was a reduction observed across all bacterial groups following anorexic media (SS2) in the proximal (V1), transverse (V2) and distal (V3) simulation. When compared to the HC-media, the levels of total bacteria (V1, V2, V3 FOS model; V2, V3 *S. boulardii* model), *Bifidobacterium* (V1, V2, V3 FOS model; V2 *S. boulardii* model), *Bacteroides* spp. (V2, V3 *S. boulardii* model; V2 FOS model), *Roseburia* (V1, V2, V3 FOS model; V2, V3 *S. boulardii* model), *Clostridium coccoides*-*Eubacterium rectale* group (V3 *S. boulardii* model), *Clostridium* cluster IX (V1, V2 FOS model; V3 *S. boulardii* model), *Faecalibacterium prausnitzii* (V1, V2, V3 FOS model), *Clostridium histolyticum* (V3 *S. boulardii* model) and *Phascolartobacterium faecium* (V2 *S. boulardii* model) following the AN-media were significantly reduced (*p*-values indicated in Figure 2). The level of *Akkermansia muciniphila* was significantly decreased following the mimicking of AN-media proximal colon in both *S. boulardii* and FOS model (5.97 to 4.59 log_10_ cells/mL, *p* < 0.05 and 5.23 to 4.73 log_10_ cells/mL, *p* < 0.05), V2 (5.80 to 5.03 log_10_ cells/mL, *p* < 0.05) and V3 (5.48 to 5.70 log_10_ cells/mL, *p* < 0.05) in FOS model.

However, following the addition of FOS to anorexic media models, a significant increase of total bacteria occurred from 8.16 to 8.42 log_10_ cells/mL (*p* < 0.05), from 7.73 to 8.28 log_10_ cells/mL (*p* < 0.05) and from 7.28 to 8.06 log_10_ cells/mL (*p* < 0.05), in V1, V2 and V3, respectively. Additionally, FOS significantly increased numbers of *Roseburia* spp. from 5.49 to 7.28 log_10_ cells/mL (*p* < 0.01), from 5.16 to 7.07 log_10_ CFU/mL (*p* < 0.01) and from 4.95 to 6.57 log_10_ cells/mL (*p* < 0.05), in V1, V2 and V3, respectively. Numbers of *Bifidobacterium* were significantly increased from 5.30 to 6.68 log_10_ cells/mL (*p* < 0.001) in V1, from 5.46 to 5.87 log_10_ cells/mL (*p* < 0.001) in V2 and from 5.46 to 5.72 log_10_ cells/mL in V3 (*p* < 0.01) after FOS administration. *Bacteroides* were significantly increased from 6.72 to 7.3 log_10_ cells/mL (*p* < 0.05) and *Phascolartobacterium faecium* were significantly increased from 5.26 to 6.04 log_10_ cells/mL in V2 (*p* < 0.05) after *S. boulardii* administration within AN-media. Administration of *S*. *boulardii* significantly stimulated growth of *Akkermansia muciniphila* from 4.59 to 5.35 log_10_ cells/mL (*p* < 0.01). FOS significantly increased numbers of *Desulfovibrio* and *Clostridium coccoides*-*Eubacterium rectale* in V1 from 4.47 to 5.62 log_10_ cells/mL (*p* < 0.05) and in V3 from 6.69 to 7.49 log_10_ cells/mL (*p* < 0.05), respectively. *S. boulardii* significantly increased numbers of *Atopobium, Clostridium histolyticum* and *Desulfovibrio* in V1 from 5.40 to 6.11 log_10_ cells/mL (*p* < 0.05), in V2 from 5.40 to 6.35 log_10_ cells/mL (*p* < 0.05) and in V3 from 4.43 to 5.28 log_10_ cells/mL (*p* < 0.05), respectively.

### 3.2. Neurotransmitter Production

Changes in neurotransmitter concentrations in gut model systems are reported in Figure 3. Compared to HC-media, AN-media led to a significant decrease in DA, NE and EPI concentrations in V1 in the proximal colon simulation. Additionally, when compared to HC-media, AN-media led to a significant decrease in EPI and NE (V2 FOS model). AN- media led 5-HT and GABA to decrease in all vessels. However, DA and EPI production were significantly increased in V1 from 1.07 to 2.97 ng/mL (*p* < 0.05) and from 2.99 to 10.48 ng/mL (*p* < 0.05) after FOS administration with AN-media. Compared to AN-media, the fermentation of *S. boulardii* mediated a significant increase in production of 5-HT (from 73.45 to 97.60 ng/mL) and GABA (from 146.61to 349.55 ng/mL) both in V1, stimulating the proximal colon (both *p* < 0.05).

### 3.3. SCFA and BCFA Production

Changes in SCFA and BCFA concentrations are shown in Figure 4. This study shows a lower concentration of acetate and butyrate following AN-media in all vessels compared to HC-media. Levels of acetate (V1 *S. boulardii* model; V3 FOS model, both *p* < 0.05), butyrate (V1, V2 FOS model, both *p* < 0.01), propionate (V2 *S. boulardii* model, *p* < 0.05) and BCFA (V2 *S. boulardii* model, *p* < 0.05) were significantly decreased in AN-media. A higher concentration of BCFA in V3 *S. boulardii* model from 5.70 to 7.06 mM (*p* = 0.57) and V1 FOS model from 1.00 to 1.71 mM (*p* = 0.59) was observed following fermentation of AN-media. Supplementation of FOS to AN gut models led to a significant increase in levels of acetate in V1, V2 and V3, simulating the proximal, transverse (both *p* < 0.05) and distal (*p* < 0.001), respectively. The fermentation of FOS mediated a significant increase in concentrations of propionate in V1 (*p* < 0.01) and butyrate in V1 and V3 (*p* < 0.01 and *p* < 0.05), simulating proximal and distal colon. FOS media led to a decrease in levels of BCFA in V1 from 1.71 to 0.31 mM (*p* = 0.27) and V2 from 3.50 to 3.03 mM (*p* = 0.76), simulating the proximal and transverse colon.

## 4. Discussion

An in vitro gut model was used to explore the impact of dietary changes, as seen in AN, on the bacterial community and metabolic end products. The subsequent impact of a pre and probiotic was assessed to determine if such an intervention could enhance microbiota and metabolites. Compared to a healthy-based medium, the results showed a limited gut microbial community and metabolite profile following nutrient-restricted conditions. However, both pre and probiotics within these AN conditions resulted in recovery of bacterial populations and key metabolites. The data presented in this paper demonstrate this novel anorexic colonic model system and further expand on understanding the impact of starvation on the microbial community. This study has confirmed that reduced nutrients are associated with profound alterations of bacterial community structure, with reductions observed in total bacteria and several specific bacterial groups. In line with this, microbial diversity and composition in recent AN studies have reported differences to healthy and overweight individuals. However, this research goes further by linking these changes to alterations in microbial metabolites that might be pertinent for mental well-being and by exploring the outcome of experimental supplementation of pre and probiotics.

In the current study ‘anorexic’ intake resulted in decreased numbers of *Bifidobacterium*, *Lactobacillus* and *Faecalibacterium prausnitzii* bacterial groups, which may protect against gut mucosal barrier function abnormalities. In animal studies, induced starvation is associated with decreased mucin production, which may lead to thinning of the mucosal layer and increased gut permeability, known as “leaky gut” [59,60]. Such a state is linked to the immune system and GBA that can trigger the release of proinflammatory cytokines leading to higher levels of inflammation, correlated with depression [61]. Those with conditions such as depression, for example, have often been reported to have increased gut permeability which can also be affected by starvation [15,62]. In the current model it is worth noting that the mucin concentrations were the same in the HC and AN models, suggesting that other factors might impact levels of these microorganisms. It is also of interest that *Akkermanisia muciniphila* has been observed to increase in AN patients, but not in the current study. This could be because in AN patients mucin starts to be a valuable substrate for gut microbiota with the reduction of several other nutrient sources in a way that was not mimicked in the current experiment.

AN studies have observed decreased numbers of *Roseburia* [63], findings supported in this in vitro study showing lower *Roseburia* and its dominant metabolites butyrate and propionate. However, a general increase in both bacterial groups after inulin and *S. boulardii* treatment was observed, suggesting that pre and probiotic administration may have a positive impact on the growth of these bacteria. SCFA are believed to have direct anti-inflammatory effects in the gut [64,65]. Decreased *Roseburia* spp. abundance has also been found in patients with inflammatory bowel diseases [66]. In this study, both interventions increased *Roseburia* spp. Higher levels of this microorganism may support reduced levels of inflammation and better gut barrier function, therefore having a positive influence on the GBA. As such, the interventions used could provide a feasible approach in AN through positively influencing the microbial community.

Compared to HC feeding, a decreased abundance of *Clostridium* spp. was observed in AN models, which coincided with decreased SCFA levels, and was in line with the findings of Borgo et al. [2], who demonstrated that AN individuals have relatively fewer carbohydrate fermenters, such as decreased numbers of *Clostridium*, which supports lower faecal butyrate concentration. Indeed, the introduction of the AN-media led not only to large reductions in the bacterial community, but also in the levels of SCFA and neurotransmitters, phenomena which have previously been observed as a result of dietary restriction [67]. So far, several studies have explored faecal excretion of SCFA in AN individuals, compared to healthy individuals, noting a reduction of faecal concentrations of mainly butyrate and propionate [2,8,13]. This is likely to be a result of the reduced fermentation capacity by the AN microbiota, characterised by lower levels of SCFA-producing bacteria and reduced carbohydrate levels, consistent with microbial signatures observed [2]. Recent studies have focused on SCFA and their effect on the CNS. These fermentation products can cross the blood-brain barrier (BBB) and might influence early brain development [68,69]. Borgo et al. [2] evaluated SCFA levels in plasma finding that acetate was the only metabolite detected, suggesting transmission across the BBB. It should be considered that SCFA act as key metabolites on peripheral tissues as a substrate for lipogenesis and acting on appetite regulation [70,71]. SCFA as signalling molecules could affect the GBA through a modulation of the ENS system, by stimulating gut hormones and cytokine release or directly via afferent neural pathways [72]. Therefore, a hypocaloric diet typically characterised by low carbohydrate intake could result in lowering faecal SCFA levels in individuals with AN, likely by developing improved mechanisms in absorption and digestion of nutrients in the gut and prolonging the colonic transit time due to constipation [67,73]. The observations of SCFA changes could be of great relevance. Compared to HC and AN feeding, as the amount of starch as well as fibre may directly correlate to butyrate, acetate and propionate levels. SCFA participate in endocrine regulation and impact on physiological and psychological functions. SCFA are involved in regulating the expression of appetite hormones such as peptide YY (PYY) and ghrelin [74]. Ghrelin is known as an appetite-stimulating hormone and germ-free mice have been observed to have significantly decreased ghrelin levels compared to conventional mice. Infusion of acetate increased both ghrelin levels and caloric intake, indicating that ghrelin expression, and thus appetite, can be enhanced through increased acetate levels [75]. As such, there are communication pathways among the gut microbiome, SCFA and anorexigenic/orexigenic hormones, that could hold potential in therapeutic feeding regimes for AN.

It is also worth considering other microbial groups that were not targeted in this study. For example, Enterobacteriaceae has been considered capable of enhanced energy extraction from the diet. Furthermore, Enterobacteriaceae has a role to play in the production of an anorexigenic bacterial protein, Caseinolytic protease B, which may be able to stimulate PYY production [76]. While this microbial group has not been a focus for the current research, it may also have a role to play in hormone secretion and satiety and thus warrants further study.

Studies have indicated that body mass index (BMI) and weight are positively related to both butyrate and propionate concentrations in stool samples [2,77], whereas butyrate levels have been observed to be negatively correlated with depression and anxiety scores. Both mice [78] and human [2] studies have indicated that behaviour disruption occurs alongside changes in gut microbiota. The current study also indicated that the introduction of pre and probiotic supplementation could positively impact on both the microbial community and SCFA levels without further modulating the diet. This is of great potential in regulating the impact of bacterial metabolic activity in the colon to improve gut health and mood.

This paper has demonstrated that five neurotransmitters in faecal fluid can successfully be measured by LC/MS. Only a few LC/MS methods have been described to determine plasma and urine neurotransmitters levels, and the uniqueness of gut model faecal supernatant neurotransmitter analysis and quantification may provide a clue about GBA connection. Whilst colonic cells are often considered as also required in the production of many neurotransmitters, following microbial fermentation in the absence on colonic cells all five neurotransmitters were detected to be at lower concentrations following AN-media compared with HC-media. 5-HT, DA and NE are classified by monoaminergic neurotransmitters and both human and animal studies stated that diet-induced starvation depletes central monoamines, leading to dysregulated neurotransmitter levels and receptor sensitivity [79,80,81]. 5-HT is synthesised from its precursor tryptophan, an essential amino acid that must be obtained through the diet [82]. The lower levels of 5-HT observed in this AN model could be due to the low protein content, since tryptophan is a necessary precursor of 5-HT. A study from Prochazkova et al. (2019) [63] found that decreased 5-HT, tryptophan and 5-HT metabolite levels in AN individuals’ stool samples compared to healthy controls, and Bailer et al. [83] suggested that AN individuals have a reduction in serotonergic activity due to reduced dietary supplies of 5-HT precursors.

DA is involved in evaluating the hedonic aspects of food in AN shown through decreased CSF concentrations of DA metabolites [84]. Both altered 5-HT and DA activity increase after recovery from AN [84,85] and, in the current study, after pre and probiotic supplementation with AN-feeding, levels of 5-HT and DA were recovered compared to AN-feeding, especially with *S. boulardii* supplementation. This could suggest that pre and probiotics that target neurotransmitters (psychobiotics) may, by supporting the microbiota, lead to the enhancement of neurotransmitter levels. The therapeutic impact of these changes remains to be determined, as the neurotransmitter levels here are lower than would be found in a host, although such changes may help to improve gut symptoms and mental health.

It is worth noting that there are several limitations associated with the use of model systems. For example, for neurotransmitter production it is often deemed necessary to have human cells present, the low levels of production that have been observed indicate the potential involvement of the microbial community in direct neurotransmitter production. In addition, whilst a limited nutrient environment has been modelled, it has not been possible to model the human secretions as they would naturally occur. To counter this, components like mucin and bile salts have been added to the media, but the amounts used may not be completely appropriate. Furthermore, AN is a multifaceted and complex condition; the idea of the model was to determine how dietary restriction might impact on the microbial community, starting from a ‘healthy’ bacterial population and the subsequent effect this might have on microbial metabolites. The addition of prebiotic and yeast added more sustenance to the media, therefore resulting in microbial growth. It could be argued that re-feeding would also result in such changes, but the approach of using pre and probiotics is designed to give a targeted response, using far fewer nutrients and resulting in positive changes to the microbial community and subsequently to neuro-metabolic related secretions. It is recognised that this study does not offer a replication of the in vivo bacterial community in AN, but can give an indication of the impact of drastic dietary changes on the microbial community and metabolites.

Following FOS feeding, positive metabolic alterations were observed from bacterial profiles and metabolite levels combined with increased abundance of 5-HT and GABA. These current findings indicate that prebiotics may be effective in the treatment of neurological problems including depression and anxiety. Under the *S. boulardii* administration, DA level were increased sharply, although no research mentioned DA levels in faeces, some studies indicated DA to be found in blood (0.03 ng/mL) [86] and urine (65,000 to 400,000 ng/24 h) [87], indicating that probiotic yeast *S. boulardii* can boost microorganism neurotransmitter production within a physiological relevant range. The current data indicate potential for use of both pre and probiotics to modulate the microbiota and also factors associated with GBA.

DA, EPI and NE are three primary catecholamines derived from the presence of tyrosine, an amino acid found in dietary proteins. DA is further degraded into EPI and NE. All three neurotransmitters play a crucial role in maintaining the GI tract, including nutrient absorption, innate immune function and gut motility [88]. For example, EPI regulates smooth muscle relaxation and colonic motor function [89]. Compared with HC feeding, the level of catecholamines in AN feeding was significantly reduced in the current study. This may be due to the lack of L-tyrosine to support catecholamine synthesis, implying that AN individuals suffer from dysregulated neurotransmitters, mood disorders and GI tract disorders. FOS administration on AN feeding had a potential effect on catecholamines production and appeared to aid its synthesis.

## 5. Conclusions

This in vitro study examined the impact of dietary restriction associated with anorexia nervosa on the gut microbial community and its metabolites. This model is not an exact replication of the in vivo bacterial community in AN, but it does highlight the close link between reduced dietary intake, bacterial community and gut homeostasis status. The pre and probiotics show potential as psychobiotics, as administration showed promising and positive results that might represent an approach to positively supporting the microbiota, metabolites and neuroactive compounds in a way that could be of benefit to an AN host.

## Figures and Tables

**Figure 1 nutrients-13-04447-f001:**
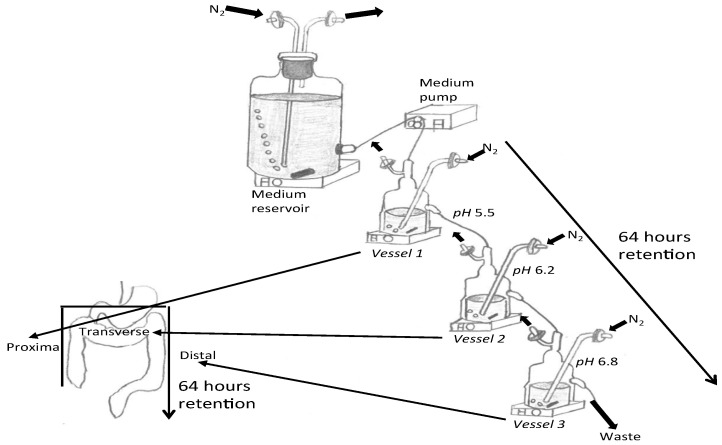
Schematic of gut model system indicating the retention time and the relation of the vessels to the large intestine.

**Figure 2 nutrients-13-04447-f002:**
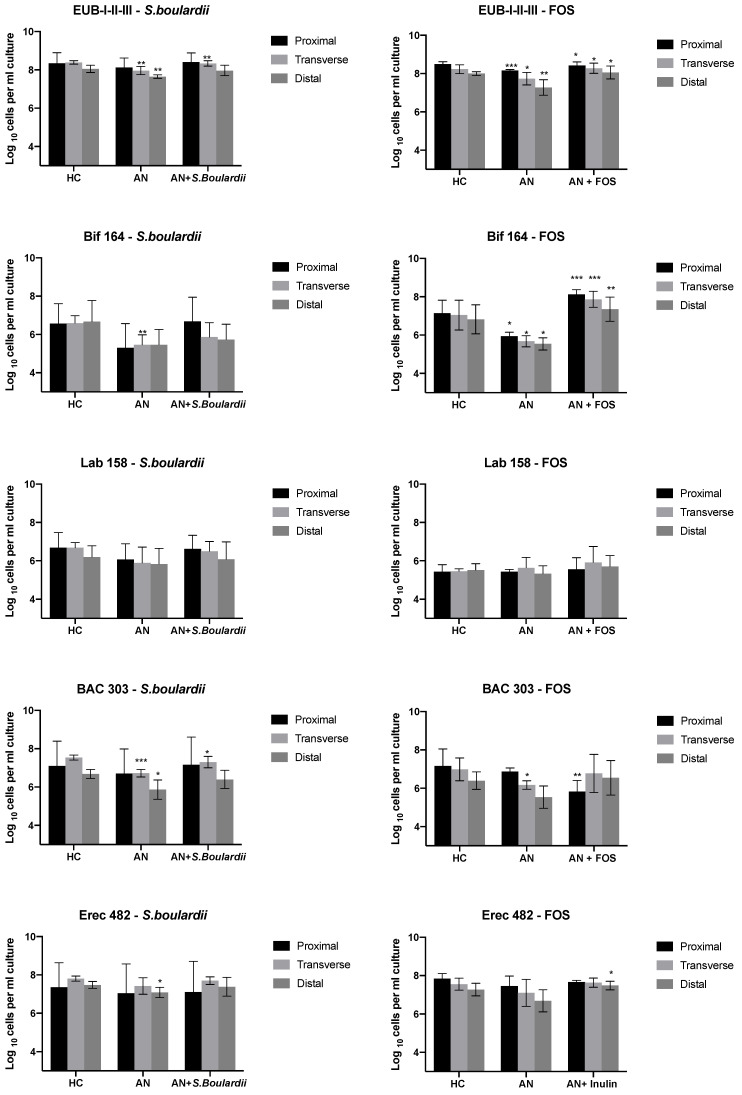
Bacterial groups at different steady states detected by FISH-FCM (Log_10_ cells/mL) from each vessel V1, V2 and V3 mimicking the proximal, transverse and distal colon of in vitro colonic model. Samples were collected at SS1 (Healthy media), SS2 (Anorexic media) and SS3 (Anorexic media with the daily administration of *Saccharomyces boulardii*/FOS). Values are mean ± SD. Significant difference in each vessel * *p* < 0.05; ** *p* < 0.01; *** *p* < 0.001 between SS1 and SS2, SS2 and SS3 are indicated.

**Figure 3 nutrients-13-04447-f003:**
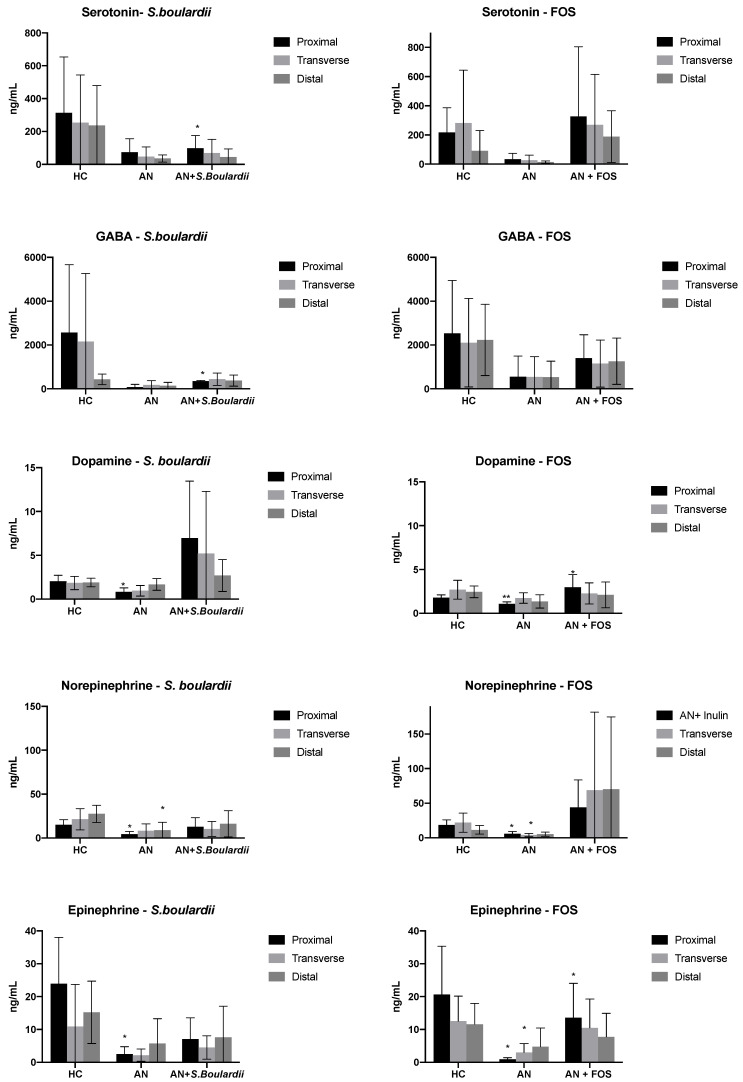
LCMS analysis. 5-HT, GABA, DA, NE and EPI concentration in culture broths with SS1 (healthy media), SS2 (anorexic media) and SS3 (anorexic media with the daily administration of *Saccharomyces μL boulardii*/FOS recovered from V1, V2 and V3) (stimulation of proximal, transverse and distal) of in vitro colonic model systems. Results are reported as Means ± SD (ng/mL). For each measurement, significant differences in each vessel * *p* < 0.05; ** *p* < 0.01 between SS1 and SS2, SS2 and SS3 are indicated.

**Figure 4 nutrients-13-04447-f004:**
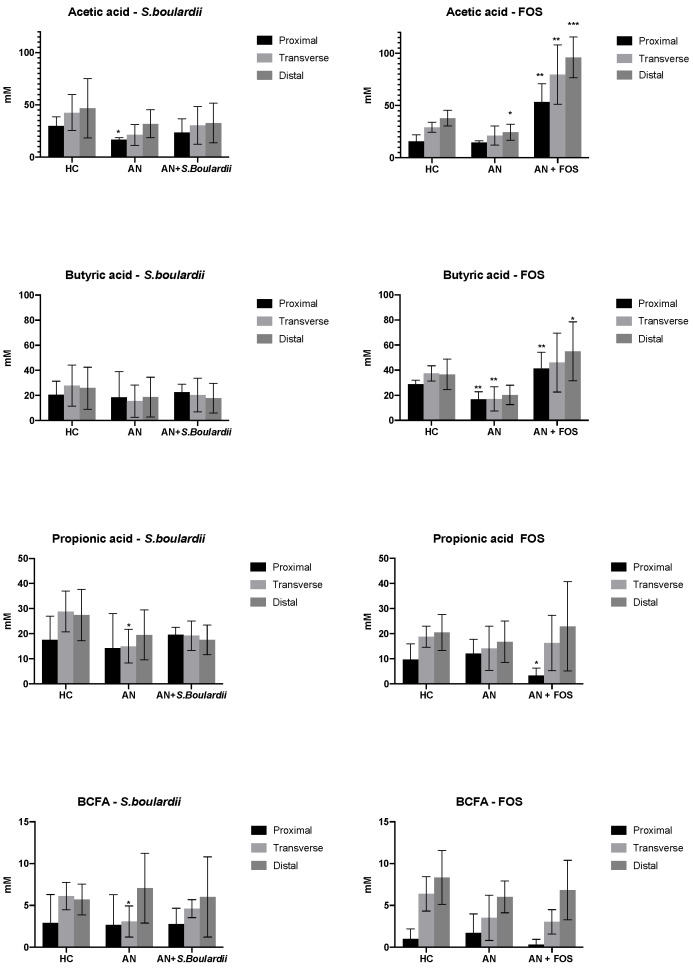
GC analysis. Acetate, propionate, butyrate and BCFA concentration in culture broths with SS1 (healthy media), SS2 (anorexic media) and SS3 (anorexic media with the daily administration of *Saccharomyces boulardii*/FOS recovered from V1, V2 and V3) (stimulation of proximal, transverse and distal) of in vitro colonic model systems. Results are reported as Means ± SD (mM). For each measurement, significant differences in each vessel * *p* < 0.05; ** *p* < 0.01; *** *p* < 0.001 between SS1 and SS2, SS2 and SS3 are indicated.

**Table 1 nutrients-13-04447-t001:** Nutrients assessment. AN/restricted gut model recipe was determined from comparing daily dietary nutrition requirements based on the British Nutrition Foundation (BNF) guideline [41]. Compared to the nutrient intake of individuals with the restricting subtype of AN, where the illness is characterised by dietary restriction in the absence of recurrent episodes of regular binge eating and purging [42]. Additionally, each medium contained: 0.8 g L-cystine HCI, 1 mL Tween 80, 4 mL resazurin solution (0.025 g/100 mL, pH 7), 4 g mucin (Porcine gastric type III), 0.4 g bile salts per litre gut model, representing human secretions determined within original gut model media.

	Nutrients	BNF level	Quartile 1 (Restricting AN n = 12)	Comparison %	Representative Medium	Healthy Gut Model Medium (g/L) SS1	Anorexic Medium (g/L) SS2 and SS3
M A C R O N U T R I E N T S	Carbohydrate (g)	260	110	42%	Starch	5 g	2.1 g
	Protein (g)	50	33	66%	Peptone water Tryptone Yeast extract Casein	5 g 5 g 4.5 g 3 g	3.3 g 3.3 g 2.97 g 1.98 g
	Dietary fibre (g)	30	12	40%	Guar Gum Inulin Pectin Arabinogalactan Xylan	1 g 1 g 2 g 2 g 2 g	0.4 g 0.4 g 0.8 g 0.8 g 0.8 g
M I C R O N U T R I E N T S	Potassium (mg)	3500	2660	76%	KCI	4.5 g	3.28 g
	Chloride (mg)	2500	1825	73%	NaCI	4.5 g	3.28 g
	Sodium (mg)	1600	1168	73%	NaHCO_3_	1.5 g	1.095 g
	Magnesium (mg)	270	227	84%	MgSO4·7H2O	1.25 g	1.05 g
	Phosphorus (mg)	550	607	110%	KH_2_PO_4_ K_2_HPO_4_	0.5 g 0.5 g	0.55 g 0.55 g
	Calcium (mg)	700	545	78%	CaCl2·6H_2_O	0.15 g	0.117 g
	Iron (mg)	14.8	7.9	53%	Hemin FeSO4.7H_2_O	0.5 g 0.005 g	0.0265 g 0.00795 g
	Vitamin K (μg)	-		68%	Vitamin K	10 μL	6.68 μL

**Table 2 nutrients-13-04447-t002:** Oligonucleotide probes used in the study for bacterial populations by fluorescent in situ hybridisation.

Probe Name	Sequence (5′ to 3′)	Target Species	Reference
Non Eub	ACTCCTACGGGAGGCAGC	Control probe complementary to EUB338	Wallner et al. (1993) [43]
Eub338I+	GCTGCCTCCCGTAGGAGT	Most Bacteria	Daims et al. (1999) [44]
Eub338II+	GCAGCCACCCGTAGGTGT	*Planctomycetales*	Daims et al. (1999) [44]
Eub338III+	GCTGCCACCCGTAGGTGT	*Verrucomicrobiales*	Daims et al. (1999) [44]
Bif164	CATCCGGCATTACCACCC	*Bifidobacterium* spp.	Langendijk et al. (1995) [45]
Lab158	GGTATTAGCAYCTGTTTCCA	*Lactobacillus* and *Enterococcus*	Harmsen et al. (1999) [46]
Bac303	CCAATGTGGGGGACCTT	Most Bacteroidaceae and Prevotellaceae, some Porphyromonadaceae	Manz et al. (1996) [47]
Erec482	GCTTCTTAGTCARGTACCG	Most of the *Clostridium coccoides-Eubacterium rectale* group *(Clostridium* cluster XIVa and XIVb)	Franks et al. (1998) [48]
Rrec584	TCAGACTTGCCGYACCGC	*Roseburia* genus	Walker et al. (2005) [49]
Ato291	GGTCGGTCTCTCAACCC	*Atopobium* cluster	Harmsen et al. (2000) [50]
Prop853	ATTGCGTTAACTCCGGCAC	*Clostridium* cluster IX	Walker et al. (2005) [49]
Fprau655	CGCCTACCTCTGCACTAC	*Feacalibacterium prausnitzii* and relatives	Hold et al. (2003) [51]
DSV687	TACGGATTTCACTCCT	*Desulfovibrio* genus	Devereux et al. (1992) [52]
Chis150	TTATGCGGTATTAATCTYCCTTT	Most of the *Clostridium histolyticum* group *(Clostridium* cluster I and II)	Franks et al. (1998) [48]
Phasco741	TCAGCGTCAGACACAGTC	*Phascolartobacterium faecium, Acidaminococcus fermentans, Succiniclasticum ruminis*	Harmsen et al. (2002) [53]
SUBU1237	CCCTCTGTTCCGACCATT	*Burkholderia* spp.	Stoffels et al. (1998) [54]
Muc1437	CCTTGCGGTTGGCTTCAGAT	*Akkermansia muciniphila*	Audie et al. (1993) [55]

**Table 3 nutrients-13-04447-t003:** Optimal conditions of LC-MS/MS used for the quantification of DA, 5-HT, NE, EPI and GABA in faecal supernatant.

Compound Name	Precursor Ion (m/z)	Product Ion (m/z)	Retention Time (min)	Classification
5-HT	177.10	160.10	10.527	Amino acid derivative
DA	154.10	91.05	8.078	Amino acid derivative
NE	170.10	152.15	4.988	Catecholamine
EPI	184.10	166.10	7.164	Catecholamine
GABA	104.10	87.05	3.690	Organic acid

Each analyte of ionisation polarity is (+).

## Data Availability

All data generated is presented in the manuscript.

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
