# Peer review of "An In Vitro Approach to Studying the Microbial Community and Impact of Pre and Probiotics under Anorexia Nervosa Related Dietary Restrictions"

_nutrients, 2021, doi:10.3390/nu13124447_

Round 1

Reviewer 1 Report

The study is conducted with scientific rigour and well described. The conclusions are of a novel quality. The study, relying on an ‘anorexic’ diet-based gut model,  suggested that bacterial- and metabolic-influenced nutritional status can shape gut microbial community and its metabolites. Even though such a model is not a replication of the in vivo bacterial community in AN, it does highlight the close link between disordered dietary intake, bacterial community and gut homeostasis status.

The study could be enriched by stating its limitations.

There are also some minor problems with double spaces throughout the text (e.g. Introduction or Conclusions section).

Author Response

Reviewer 1

The study is conducted with scientific rigour and well described. The conclusions are of a novel quality. The study, relying on an ‘anorexic’ diet-based gut model,  suggested that bacterial- and metabolic-influenced nutritional status can shape gut microbial community and its metabolites. Even though such a model is not a replication of the in vivo bacterial community in AN, it does highlight the close link between disordered dietary intake, bacterial community and gut homeostasis status.

The study could be enriched by stating its limitations.

There are also some minor problems with double spaces throughout the text (e.g. Introduction or Conclusions section).

Many thanks for your comments. We have added a paragraph on limitations (p. 14) and have been through the manuscript to check the spacing.  We have, respectfully, included some of the reviewer’s wording regarding the limitations.

It is worth noting that there are several limitations associated with the use of model systems. For example, for neurotransmitter production it is often deemed necessary to have human cells present, the low levels of production that have been observed indicate the potential involvement of the microbial community in direct neurotransmitter production. In addition, whilst a limited nutrient environment has been modelled, it has not been possible to model the human secretions as they would naturally occur. To counter this, components like mucin and bile salts have been added to the media, but the amounts used may not be completely appropriate. Furthermore, AN is a multifaceted and complex condition; the idea of the model is to determine how dietary restriction might impact on the microbial community, starting from a ‘healthy’ bacterial population and the subsequent effect this might have on microbial metabolites. Addition of prebiotic and yeast added more sustenance to the media and therefore resulting in microbial growth. It could be argued that re-feeding would also result in such changes, but the approach of using pre and probiotics is designed to give a targeted response, using far fewer nutrients and resulting in positive changes to the microbial community and subsequently to neuro-metabolic related secretions. It is recognised that this study does not offer a replication of the in vivo bacterial community in AN, but can give an indication of the impact of drastic dietary changes on the microbial community and metabolites.

Reviewer 2 Report

Liu et al analyze fecal samples from healthy individuals after incubation in vitro in 3 different pH conditions and with adequate and deficient nutrient supply. Composition of some bacterial taxa was studied by FISH and content of some metabolites including GABA, monoamines, SCFA and BCFA have been quantified. Effects of a probiotic yeast (S. boulardii) and prebiotic FOS on these parameters have been studied. General decrease in all bacterial taxa and metabolites have been found in nutrient deficient conditions while supplementation in pre-and probiotics showed partial recovery. Authors propose to call they approach as an in vitro model to study gut microbiota (GM) in anorexia nervosa (AN). Although the experiments technically sound, the general novelty of the study and its significance to the field of GM in AN is questioned. Importantly, to validate this model, there is no data showing that in the gut of AN patients bacteria grow in the nutrient depleted state.

  1. While having an in vitro model of AN is a good idea, the present approach does not appear to answer the complexity of the in vivo situation. Moreover, from the basic biology known effects of nutrient supply on bacterial growth, there is a wealth of data in earlier microbiology papers. Authors should carefully examine and cite them. Presently, I do not see what is new about to find reduced bacterial growth and metabolism when using nutrient starvation medium.
  2. Authors cultivate only bacteria from fecal samples of healthy individuals, why you did not compare such approach with AN fecal samples. Presently, the study only evaluates effects of nutrient starvation on “healthy MB”.
  3. In the present model nutrient supply is the only source of nutrients for bacteria. In AN, gut bacteria are fed from the host body, and hence, maintain their population independently from reduced food intake. Accordingly, total reduction of bacterial content in the gut is not consistently found in AN patients. Can authors cite any study analyzing the nutrient content inside the different parts of the gut of AN patients to justify their un vitro protocol.
  4. Host starvation associated changes of GM is not sufficient to trigger disease such as AN, example of dieting in overweight and obesity which is typically does not result in AN.
  5. AN pathophysiology also involves an autoimmune component; it could be related to some gut bacterial antigens, such as ClpB protein from Enterobacteriaceae (Fetissov & Hokfelt 2019). Why Enterobacteriaceae have not been included in the list (Table 2).
  6. Adding fibers and yeasts to starving bacteria will obviously supplement them with nutrients and increase their number. Do you think that supplementation with these products makes difference from the standard refeeding strategy in AN patients, i.e. should be included. Any clinical study to cite?
  7. Authors may add a schematic figure with experimental design to better illustrate different incubation conditions which they refer as relevant to AN, presently, different “retention time” is not so clear.
  8. I would recommend do not include AN in the title and to tone down the conclusion to the modeling effects of nutrient starvation on complex bacterial population from human GM.

Round 2

Reviewer 2 Report

Authors provided a satisfactory revision, they should update the reference list to include new citations.

Author Response

Many thanks for reviewing our paper and for your additional comment; the new references have been inserted into the reference list (1 and 76).